# A Comparative Study on Oxidation of Acidic Red 18 by Persulfate with Ferrous and Ferric Ions

## Xin Li, Lijing Yuan and Liangfu Zhao *

Institute of Coal Chemistry, Chinese Academy of Sciences, Taiyuan 030001, China; lixin@sxicc.ac.cn (X.L.); yuanlijing@sxicc.ac.cn (L.Y.)

* Correspondence: lfzhao@sxicc.ac.cn; Tel.: +86-134-8536-2851

**Abstract:** Ferrous and ferric salts were tested for the persulfate activation ($PS/Fe^{2+}$ and $PS/Fe^{3+}$) and the oxidation of Acid Red 18 (AR18). A complete removal was attained after 90 min in both $PS/Fe^{2+}$ and $PS/Fe^{3+}$ processes with the persulfate concentration of 6 mM. High concentrations of PS, $Fe^{2+}$, and $Fe^{3+}$ promoted the AR18 degradation in both processes and the optimized pH were 3 and 3.3 for $PS/Fe^{2+}$ and $PS/Fe^{3+}$ processes, respectively. The mechanism of PS activation by $Fe^{3+}$ was also investigated. It was found that hydroxyl radical (HO•) and sulfate radical ($SO_4^-•$) were formed and acted as dominating radicals in both processes. It is also deduced that Fe recycle offers $Fe^{2+}$ for PS activation in $PS/Fe^{3+}$ process to produce HO• and $SO_4^-•$. The less radical side reactions lead to a higher contribution of HO• and $SO_4^-•$ on AR18 degradation in $PS/Fe^{3+}$ process.

**Keywords:** persulfate; iron; acidic red 18; activation; radicals

## 1. Introduction

Synthetic complex organic dyes are important coloring materials in textile industry. There are more than 0.7 million tons of dyes produced annually [1]. The azo dyes with nitrogen-to-nitrogen double bonds (–N=N–) are widely used in textile industry for their high chemical stability and versatility. As most azo dyes are nonbiodegradable and toxic [2,3], the dye wastewaters cannot be effectively treated by conventional biotreatment processes [4]. The presence of azo dye in surface water and groundwater may cause serious environmental problems due to the possible carcinogenicity and biotoxicity [5].

Advanced oxidation processes (AOPs) characterized by the formation of hydroxyl radicals (HO•) are effective methods for refractory organic wastewater treatment due to the strong and nonselective oxidizability of HO• [6,7]. Fenton reagent ($H_2O_2/Fe^{2+}$) oxidation is one of the most widely used AOPs for wastewater treatment since it is easy to handle and the reaction condition is relatively mild [8,9]. The reaction of $H_2O_2$ with $Fe^{2+}$ could form a large amount of HO• and thus degrade contaminants rapidly.

Persulfate ($S_2O_8^{2-}$, PS) with an O-O bond has a higher redox potential ($E^0 = 2.01$ V) than that of $H_2O_2$ ($E^0 = 1.76$ V). PS could be activated in certain conditions to yield high-reactivity radicals such as $SO_4^-•$ ($E^0 = 2.5–3.1$ V) and HO• ($E^0 = 2.80$ V) [10–12]. Activated PS oxidation methods are newly emerging AOPs for organics degradation as a result of their strong oxidizability and low material cost [12–14]. UV [15], heat [16], metal ions [17], alkali [18], and microwaves [19] are currently applied for PS activation. Since iron catalytic process is easy to operate and has a high oxidation efficiency, $Fe^{2+}$ is the most widely used transition metal ion for initializing radical chain reaction in PS activation process [14,20–22]. The reactions of PS with $Fe^{2+}$ and organics (RH) are shown as Equations (1) to (4) [14,23–25].

$$S_2O_8^{2-} + Fe^{2+} \rightarrow Fe^{3+} + SO_4^{2-} + SO_4^-•, \qquad k = 27 \text{ M}^{-1}\text{s}^{-1} \qquad (1)$$

$$SO_4^-\bullet + H_2O \rightarrow HSO_4^- + HO\bullet, \qquad k = 2.0 \times 10^3 \text{ M}^{-1}\text{s}^{-1} \tag{2}$$

$$HO\bullet + HO\bullet \rightarrow H_2O_2, \qquad k = 5.2 \times 10^9 \text{ M}^{-1}\text{s}^{-1} \tag{3}$$

$$RH + HO\bullet \text{ or } SO_4^-\bullet \rightarrow H_2O + R\bullet, \qquad k = 10^7 - 10^{10} \text{ M}^{-1}\text{s}^{-1} \tag{4}$$

$$H_2O_2 + Fe^{3+} \rightarrow Fe^{2+} + HO_2\bullet + H^+, \qquad k = 3.1 \times 10^{-3} \text{ M}^{-1}\text{s}^{-1} \tag{5}$$

$$R\bullet + Fe^{3+} \rightarrow R^+ + Fe^{2+} \tag{6}$$

$$HO_2\bullet + HO_2\bullet \rightarrow H_2O_2 + O_2, \qquad k = 8.3 \times 10^5 \text{ M}^{-1}\text{s}^{-1} \tag{7}$$

$$HO\bullet + S_2O_8^{2-} \rightarrow OH^- + S_2O_8\bullet^-, \qquad k = 1.2 \times 10^7 \text{ M}^{-1}\text{s}^{-1} \tag{8}$$

$$R\bullet \text{ or } R^+ + HO\bullet \text{ or } SO_4\bullet^- \rightarrow \text{products} \tag{9}$$

Due to the high reactivity of $Fe^{2+}$, oxidants (such as $SO_4\bullet^-$ and $HO\bullet$) would be consumed by excess $Fe^{2+}$ before they react with contaminants. Several methods have been proposed to control the release-rate of $Fe^{2+}$ and thus improve the oxidation efficiency. Chelating agents (i.e., citric acid) were used by Liang et al. [20] to combine with $Fe^{2+}$ for controlling the reaction rate of PS with $Fe^{2+}$. Oh et al. [26] and Xu et al. [27] demonstrated that $Fe^{2+}$ could be slowly released by using zero-valent iron, and found that the contaminant degradation efficiency was increased significantly.

The high reaction rates of $Fe^{2+}$ and oxidants make PS/$Fe^{2+}$ process transfer to PS/$Fe^{3+}$ process rapidly. Photocatalysis with UV or solar light was an effective method for wastewater treatment [28], and has been used for improvement of the utilization of $Fe^{2+}$ [29]. However, $Fe^{3+}$, the oxidative product of $Fe^{2+}$, is also an effective catalysis for PS oxidation process as a result of the Fe recycle [23] (Equations (5) and (6)). George and Dionysios [30] investigated the activation processes of several peroxides with nine transition metals as activators and found that both $Fe^{2+}$ and $Fe^{3+}$ were effective for PS and $H_2O_2$. As the prices of ferric salts are lower than those of ferrous salts, the reagent cost could be reduced by replacing ferrous salts with ferric salts in PS oxidation process. However, little study has been reported about the mechanism and the kinetics of contaminant oxidation by PS/$Fe^{3+}$ process under different operation conditions. The role of $Fe^{3+}$ on the PS activation and pollutions degradation remains enigmatic.

In this work, PS, combined with ferrous salt (PS/$Fe^{2+}$) and ferric salt (PS/$Fe^{3+}$), was used for the degradation of azo dye Acid Red 18 (AR18), and the oxidation abilities of PS/$Fe^{2+}$ and PS/$Fe^{3+}$ were also explored and compared. The effects of different operation conditions on AR18 degradation in PS/$Fe^{2+}$ and PS/$Fe^{3+}$ processes were investigated. The reactive species and oxidation efficiency of PS/$Fe^{2+}$ and PS/$Fe^{3+}$ processes were also compared with AR18 as model contaminants in contrast experiments.

## 2. Results

### 2.1. Comparison of PS/Fe²⁺ and PS/Fe³⁺ Processes

Figure 1 shows the comparison of different oxidation processes applied in the destruction of AR18. As can be seen in Figure 1, AR18 was difficult to be oxidized by PS alone and the addition of $Fe^{2+}$ and $Fe^{3+}$ significantly promoted the AR18 degradation. The oxidation of AR18 by PS/$Fe^{2+}$ and PS/$Fe^{3+}$ processes were represented by the following pseudo-first-order reaction kinetics

$$dC_{AR}/dt = -kC_{AR}, \tag{10}$$

where $C_{AR}$ is the AR18 concentration (mg/L), k is the reaction rate coefficient, and t is the reaction time. Equation (10), after integration, becomes

$$C_{AR} = C_{AR0} \exp(-kt), \tag{11}$$

where $C_{AR0}$ is the initial concentration of AR18.

The high correlation coefficient ($R^2$ = 0.92 and 0.99 for PS/$Fe^{2+}$ and PS/$Fe^{3+}$ processes, respectively) obtained from the linear plots of $\ln(C_{AR}/C_{AR0})$ vs. time (Figure 1) showed that the experimental date fits the pseudo-first-order model well. The reaction rate constants of PS/$Fe^{2+}$ and PS/$Fe^{3+}$ processes were 0.264 $min^{-1}$ and 0.05 $min^{-1}$, respectively. Though the reaction rate of PS/$Fe^{3+}$ process was lower than that of PS/$Fe^{2+}$ process, nearly 100% removal of AR18 was attained within 60 min in the PS/$Fe^{3+}$ process.

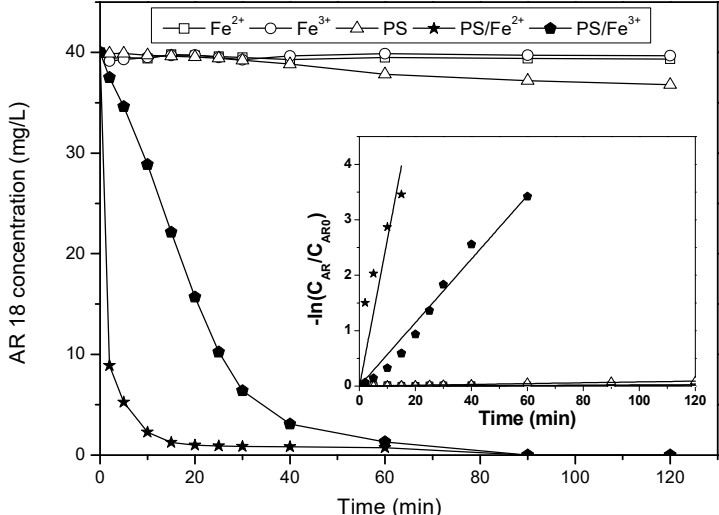

**Figure 1.** AR18 degradation curves by different oxidation processes (pH = 4.5, $C_{Fe2+}$ = $C_{Fe3+}$ = 0.1 mM, $C_{PS}$ = 3 mM, $C_{AR0}$ = 40 mg/L).

## 2.2. Effect of the Initial pH

Figure 2 illustrates the effects of the initial pH on AR18 degradation in PS/$Fe^{2+}$ and PS/$Fe^{3+}$ processes. It can be seen that an increasing initial pH benefited the AR18 degradation in both PS/$Fe^{2+}$ and PS/$Fe^{3+}$ processes when the initial pH was relatively low (pH < 3.3 in PS/$Fe^{2+}$ process and pH < 3 in PS/$Fe^{3+}$ process). The residual AR18 concentration decreased from 49 mg/L to 4 mg/L and 59 mg/L to 15.6 mg/L in 30 min with an increasing initial pH from 1.6 to the suitable values (3.3 for PS/$Fe^{2+}$ process and 3 in PS/$Fe^{3+}$ process) for PS/$Fe^{2+}$ and PS/$Fe^{3+}$ processes, respectively. When the initial pH further increased, the residual AR18 concentration increased. Radicals are important oxidants in activated PS oxidation process and the pH greatly affected the formation of radicals. It has been reported that $SO_4\bullet^-$ is the main radical in PS activation process when the pH was lower than 7 [31]. An increasing pH could avoid the formation of $(Fe(II)(H_2O))^{2+}$ and inhibit acid-catalyzed PS breakdown [32,33] (Equations (12) and (13)). The former would promote the AR18 degradation due to the enhancement of the reaction of $Fe^{2+}$ with PS, while the latter is unfavorable for AR18 degradation due to the inhibition of $SO_4\bullet^-$ formation. A suitable pH was important for both PS/$Fe^{2+}$ and PS/$Fe^{3+}$ processes. The residual AR18 concentration reached a valley value and the reaction kinetic constant reached the peak value with the initial pH of 3 in PS/$Fe^{3+}$ process and pH of 3.3 in PS/$Fe^{2+}$ process. A similar conclusion could be found in previous studies on $H_2O_2$/ $Fe^{2+}$ systems [22]. Though Ferrous salts can be dissolved under a pH of 2–9, the ferric oxyhydroxides may be generated under a higher pH (>3.0) [34]. A higher pH would lead to a loss of soluble iron ions through the precipitation process, and result in a decrease of the AR18 degradation rate. It is necessary to balance the ferric oxyhydroxides formation and acid-catalyzed PS breakdown. Therefore, the initial pH of 3.3 and 3 were the suitable pH values for PS/$Fe^{2+}$ and PS/$Fe^{3+}$ processes, respectively.

$$S_2O_8^{2-} + H^+ \rightarrow HS_2O_8^-, \tag{12}$$

$$HS_2O_8^- \rightarrow H^+ + SO_4\bullet^- + SO_4^{2-}, \tag{13}$$

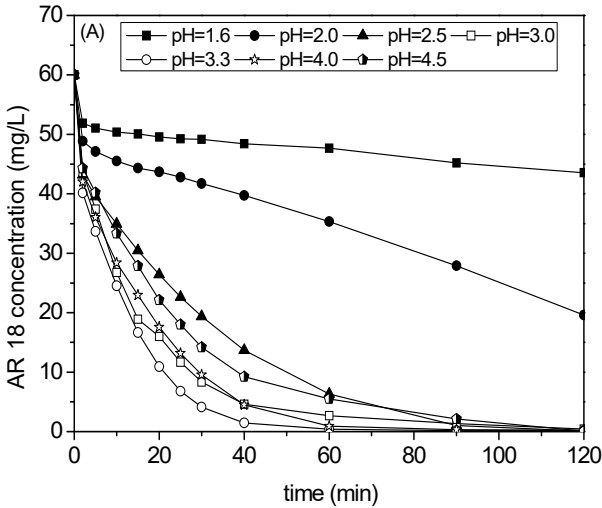

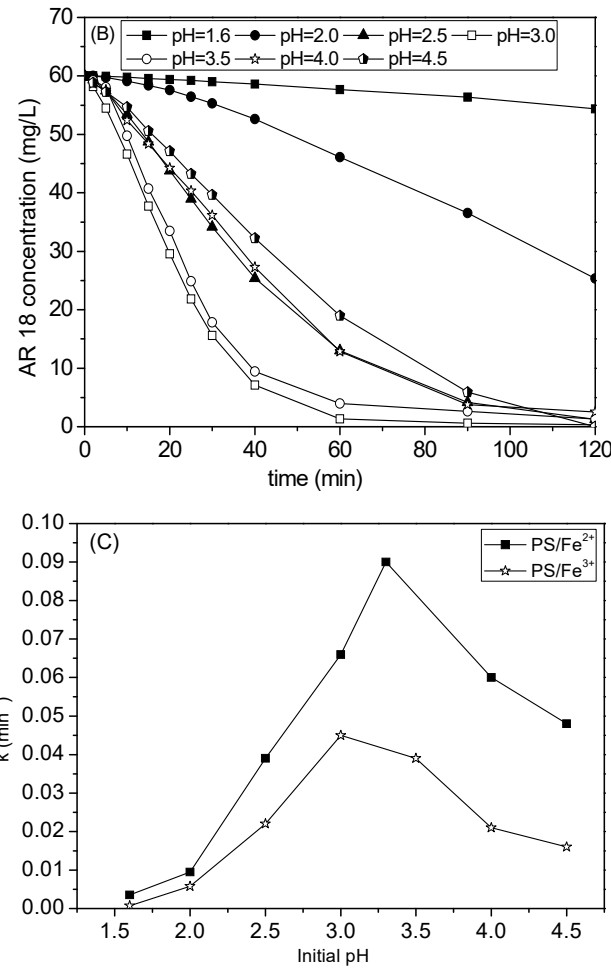

**Figure 2.** Effect of initial pH on AR18 degradation in PS/Fe$^{2+}$ (**A**) and PS/Fe$^{3+}$ (**B**) processes and the variations of reaction kinetic constants (**C**) (C$_{Fe2+}$ = C$_{Fe3+}$ = 0.1 mM, C$_{PS}$ = 3 mM, C$_{AR0}$ = 60 mg/L).

### 2.3. Effect of $Fe^{2+}$ or $Fe^{3+}$ Concentration

The influences of $Fe^{2+}$ or $Fe^{3+}$ concentration on AR18 degradation in PS/$Fe^{2+}$ and PS/$Fe^{3+}$ processes are shown in Figure 3. It is observed that AR18 cannot be effectively oxidized by sole PS ($C_{Fe2+}$ = $C_{Fe3+}$ = 0 mM). The addition of $Fe^{2+}$ or $Fe^{3+}$ significantly improved the oxidizability of PS and the AR18 degradation rate increased with the increasing $Fe^{2+}$ or $Fe^{3+}$ concentration in both PS/$Fe^{2+}$ and PS/$Fe^{3+}$ processes. It also can be seen from Figure 3C that the reaction kinetic constants increased from 0.04 min$^{-1}$ to 0.4 min$^{-1}$ and from 0.02 min$^{-1}$ to 0.115 min$^{-1}$ with an increasing $Fe^{2+}$ or $Fe^{3+}$ concentration of 0 mM to 0.5 mM in PS/$Fe^{2+}$ and PS/$Fe^{3+}$ processes, respectively. An increasing $Fe^{3+}$ concentration improved the recycle of Fe (Equation (4)) to offer more $Fe^{2+}$ for radical chain reaction. A relatively high concentration of $Fe^{2+}$ enhanced the reaction of PS with $Fe^{2+}$ (Equation (1)) to form more radicals, which resulted in the increase of AR18 degradation rate. It is noted from Figure 3B that only a slight increase in AR18 degradation rate was attained when the $Fe^{3+}$ concentration increased from 0.3 mM to 0.5 mM. This may be ascribed to the concentration of reducing organic radicals (R•) being much lower than $Fe^{3+}$ concentration, and the low concentration of R• limited the rate of Fe recycle reaction (Equation (4)).

### 2.4. Effect of PS Concentration

PS is the main source of radicals and one of the important oxidants in PS/$Fe^{2+}$ and PS/$Fe^{3+}$ processes. The influence of PS concentration on AR18 degradation was investigated by varying the PS concentration from 1 mM to 6 mM.

Figure 4 presents the effects of the PS concentration on AR 18 degradation. The results indicate that the higher PS concentration benefits the AR18 degradation in PS/$Fe^{2+}$ and PS/$Fe^{3+}$ processes. When the PS concentration increased from 1 mM to 6 mM, the AR18 concentration decreased from 10.9 mg/L (degradation ratio 82%) to 1.3 mg/L (degradation ratio 98%) and from 27.8 mg/L (degradation ratio 54%) to 0.65 mg/L (degradation ratio 98.9%) within 90 min in PS/$Fe^{2+}$ and PS/$Fe^{3+}$ processes, respectively. The result of reaction kinetic research also showed a similar trend (Figure 4C). High PS concentration increased the reaction rate of PS with Fe ion to generate more radicals, resulting in more AR18 degradation, which was also observed by other researchers [14].

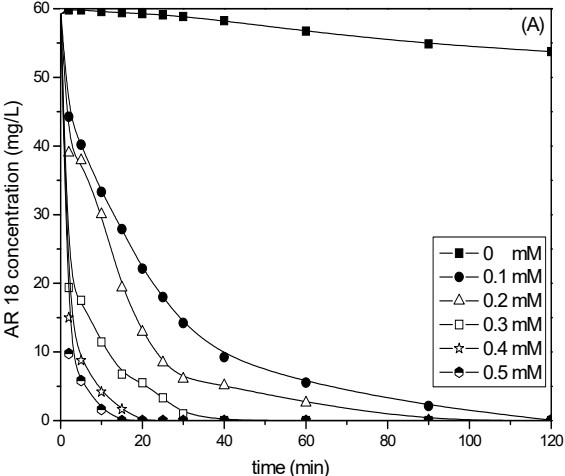

**Figure 3.** *Cont.*

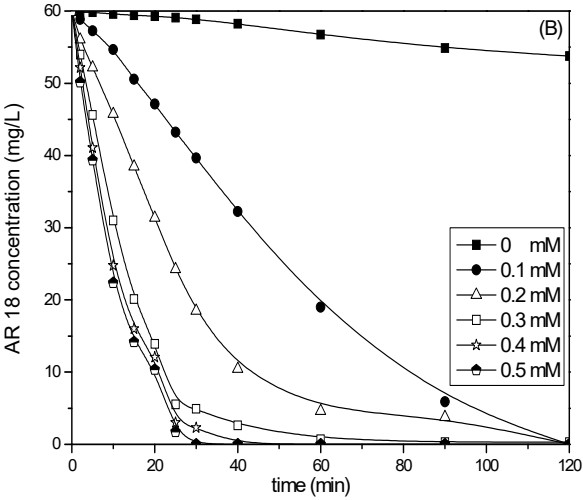

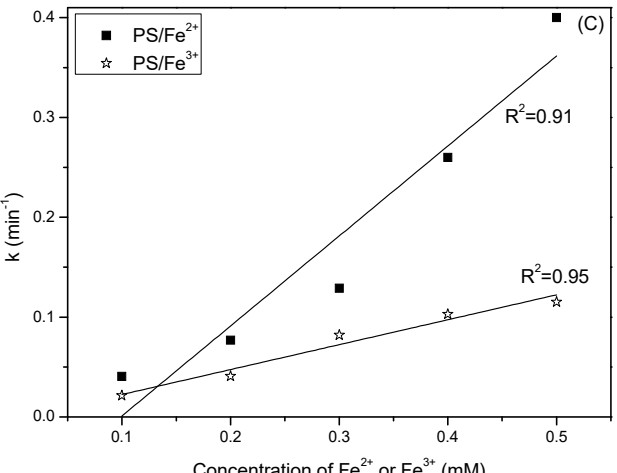

**Figure 3.** Effect of Fe ion concentrations on AR18 degradation in PS/Fe$^{2+}$ (**A**) and PS/Fe$^{3+}$ (**B**) processes and the variations of reaction kinetic constants (**C**) (initial pH = 4.5, $C_{PS}$ = 3 mM, $C_{AR0}$ = 60 mg/L).

## *2.5. AR18 Mineralization and Radical Detection*

### 2.5.1. Mineralization of AR18

Contrast experiments were carried out to compare the AR18 degradation in PS/Fe$^{2+}$ and PS/Fe$^{3+}$ processes. As shown in Figure 5, the AR18 degradation with PS/Fe$^{2+}$ occurred more rapidly than that with PS/Fe$^{3+}$ and almost 100% AR18 degradation was attained within 40 min in both PS/Fe$^{2+}$ and PS/Fe$^{3+}$ processes. The low recycle rate of Fe would be a probable reason of the lower AR18 and total organic carbon (TOC) degradation rates in PS/Fe$^{3+}$ process. After 120 min of oxidation in PS/Fe$^{2+}$ and PS/Fe$^{3+}$ processes, TOC removal ratios of about 20% were attained with AR18 completely removed. The reason for this is that the intermediates of AR18 degradation were more recalcitrant and not easy to be oxidized. Xu et al. [21] found that azo dye could be degraded more completely in PS/Fe$^{2+}$ than in H$_2$O$_2$/Fe$^{2+}$ process due to the longer half-life of sulfate radicals. Therefore, the PS/Fe$^{2+}$ and PS/Fe$^{3+}$ processes are still an effective method for organic degradation due to the high oxidation ability of SO$_4^-\bullet$.

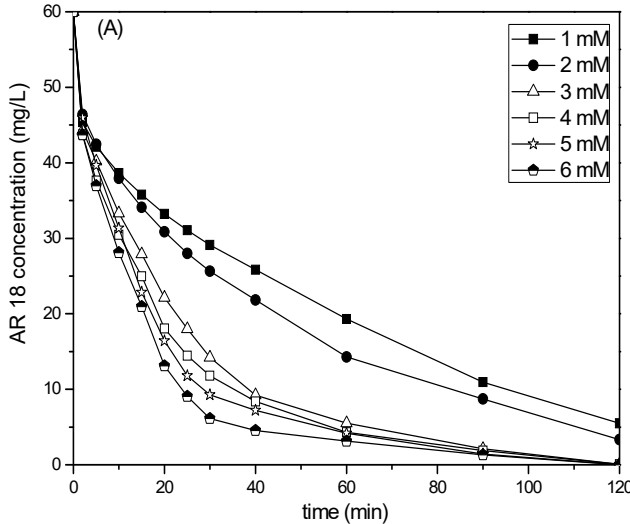

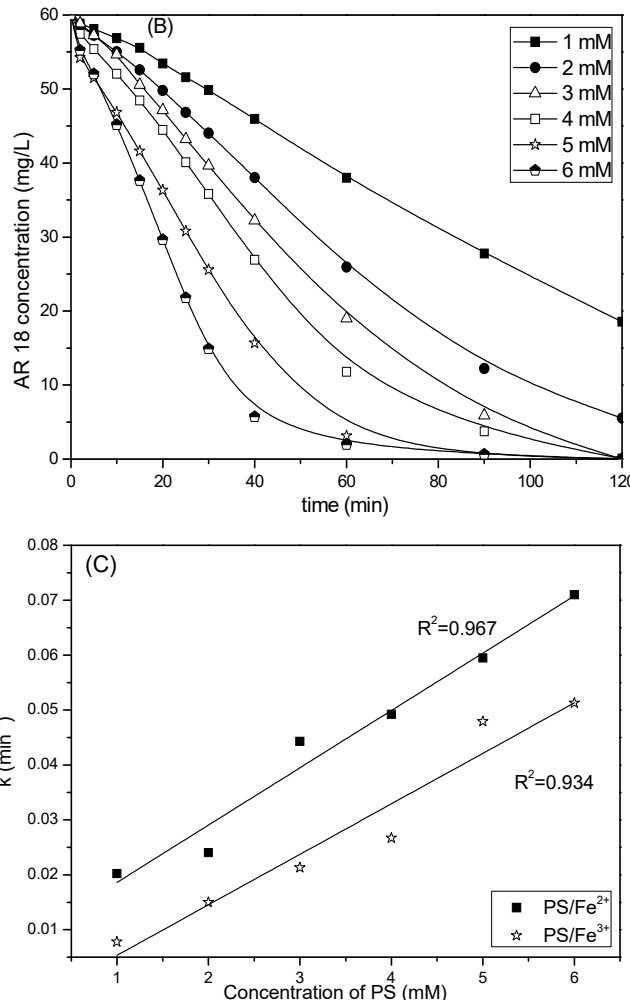

**Figure 4.** Effect of PS concentrations on AR18 degradation in PS/Fe$^{2+}$ (**A**) and PS/Fe$^{3+}$ (**B**) processes and the variations of reaction kinetic constants (**C**) (initial pH = 4.5, $C_{Fe2+} = C_{Fe3+} = 0.1$ mM, $C_{AR0} = 60$ mg/L).

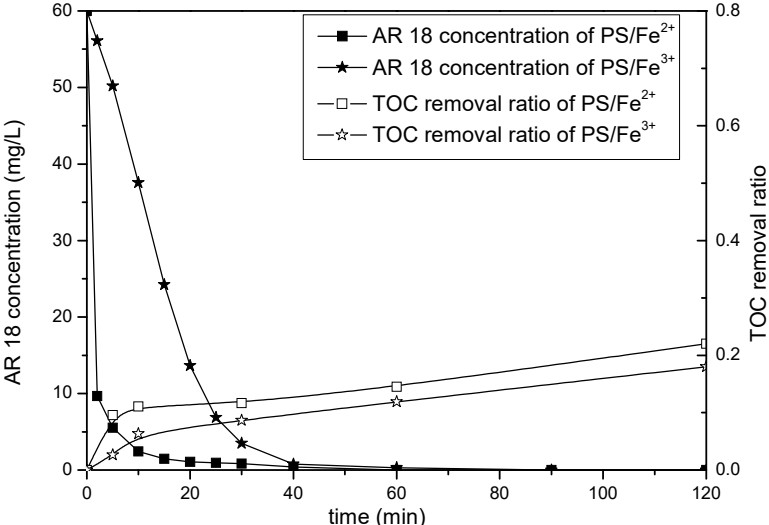

**Figure 5.** Comparison of the AR18 and total organic carbon (TOC) degradation in PS/Fe$^{2+}$ and PS/Fe$^{3+}$ processes (initial pH = 3, $C_{Fe2+}$ = $C_{Fe3+}$ = 0.1 mM, $C_{PS}$ = 6 mM, $C_{AR0}$ = 60 mg/L).

### 2.5.2. EPR Radical Detection and AR18 Degradation Mechanism

In order to investigate the PS activation mechanism, the radicals formed in PS/Fe$^{2+}$ and PS/Fe$^{3+}$ processes were analyzed by using electron paramagnetic resonance (EPR) spectroscopy with DMPO as a spin trap agent. DMPO could react with different radicals in oxidation process to form unique radical adducts. The radical adducts were much more stable than radicals (e.g., hydroxyl radical, sulfate radical) and can be determined by using an EPR spectroscopy.

The standard spectra of DMPO-hydroxyl radical (DMPO-OH, hyperfine splitting constants of AN = AH = 15 G) and DMPO-sulfate radicals (DMPO-SO$_4$, hyperfine splitting constants of AN = 13.51 G, AH = 9.93 G, $A_{\gamma1}$H = 1.34 G, and $A_{\gamma2}$H = 0.88 G) were simulated by using Isotropic Radicals software and are presented in Figure 6A. The EPR spectra of PS/Fe$^{2+}$ and PS/Fe$^{3+}$ processes are shown in Figure 6B. The EPR spectrum of DMPO-OH mixed with DMPO-SO$_4$ (ratio of 2:1) was simulated and is also shown in Figure 6B. It can be seen from Figure 6B that the EPR spectrum of PS/Fe$^{3+}$ process was similar with that of PS/Fe$^{2+}$ process and the simulation spectrum fitted well with the EPR spectra of PS/Fe$^{2+}$ and PS/Fe$^{3+}$ processes. This result indicates that the catalytic mechanism of PS activation by Fe$^{3+}$ is similar to that of PS/Fe$^{2+}$ and HO$\bullet$ and SO$_4^-\bullet$ formed in both PS/Fe$^{2+}$ and PS/Fe$^{3+}$ processes. According to the research of Liang et al. [34], SO$_4^-\bullet$ was the dominant radical at low pH and the concentration of HO$\bullet$ was about 5% of SO$_4^-\bullet$ at pH of 4. However, only a weak signal of DMPO-SO$_4$ was determined in Figure 6B. The reason for this may be the high reaction rate of DMPO with HO$\bullet$ ($3.4 \times 10^9$ M$^{-1}$s$^{-1}$) [35] and the conversion of DMPO-SO$_4$ to DMPO-OH through a nucleophilic substitution [36].

According to the EPR studies, the possible AR18 degradation mechanism in PS/Fe$^{3+}$ process was proposed and shown in Figure 7. Firstly, AR18 (RH) reacted with radicals (formed by PS decomposition) and yielded R$\bullet$. Fe$^{2+}$ was generated through the reaction of Fe$^{3+}$ with R$\bullet$. Subsequently, PS was activated by Fe$^{2+}$ to form SO$_4^-\bullet$ and then generate HO$\bullet$ and H$_2$O$_2$ (Equations (1)–(3)). Fe$^{2+}$ could regenerate through the reaction of Fe$^{3+}$ and R$\bullet$ or H$_2$O$_2$. The HO$\bullet$ also can be yielded through the reaction of H$_2$O$_2$ with Fe$^{2+}$ (Fenton reaction). With a large amount of radicals formed, AR18 can be oxidized by radicals to intermediates and then CO$_2$ and H$_2$O.

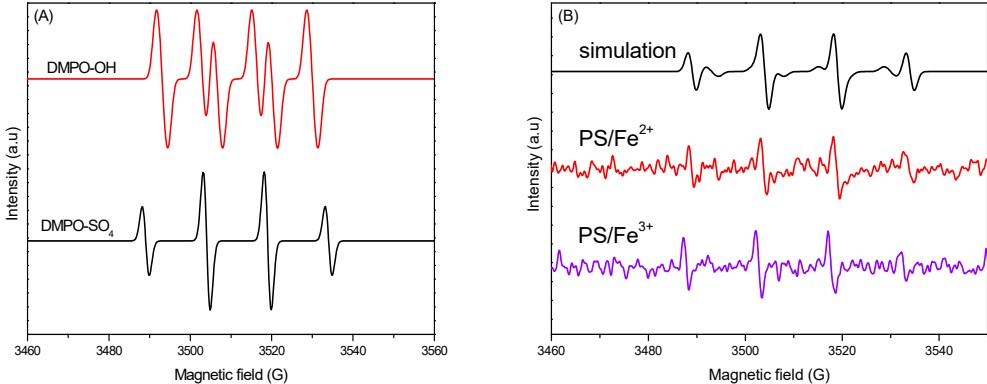

**Figure 6.** EPR spectra of DMPO-radical adducts: (**A**) standard spectra of DMPO-OH and DMPO-SO$_4$; (**B**) spectra of simulation of DMPO-OH mixed with DMPO-SO$_4$ (ratio of 2:1), PS/Fe$^{2+}$, and PS/Fe$^{3+}$.

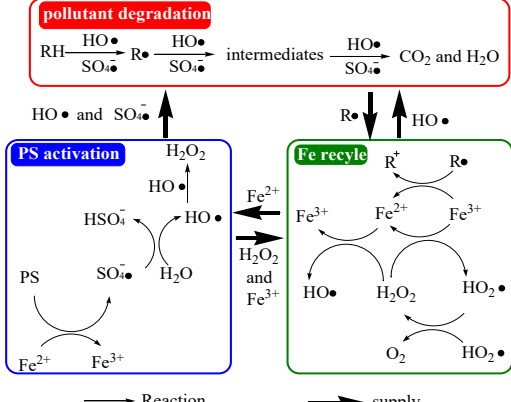

**Figure 7.** Proposed AR18 (RH) degradation mechanism in PS/Fe$^{3+}$ process.

### 2.5.3. Quenching Experiments

For a further study of the roles of different radicals, quenching experiments of AR18 degradation in PS/Fe$^{2+}$ and PS/Fe$^{3+}$ processes were carried out with ethanol (EtOH) and nitrobenzene (NB) as radical scavengers. The reaction rate constant of NB with HO• ($3.9 \times 10^9$ M$^{-1}$s$^{-1}$) is about 1000 times higher than that of NB with SO$_4^-$• ($< 10^6$ M$^{-1}$s$^{-1}$). EtOH reacts rapidly with both HO• and SO$_4^-$•. Therefore, excessive NB and EtOH were added to mainly scavenge HO• and both HO• and SO$_4^-$•, respectively. The contributions of different radicals on oxidation could be determined by comparing the AR18 degradation ratios after adding excessive NB or EtOH. This method for identification of HO• and SO$_4^-$• has been used in a great deal of research [14,37].

The effects of different radical scavengers (EtOH and NB) on AR18 degradation in PS/Fe$^{2+}$ and PS/Fe$^{3+}$ processes are presented in Figure 8A,B. It can be seen that the additions of EtOH and NB were unfavorable for AR18 degradation and the inhibiting effect of EtOH was stronger than that of NB. It can be revealed that both HO• and SO$_4^-$• are important for AR18 degradation. Based on the data of quenching experiments, the contributions of different oxidants for AR18 degradation in PS/Fe$^{2+}$ and PS/Fe$^{3+}$ processes were determined and are shown in Figure 8C. It can be seen that the contributions of HO• and SO$_4^-$• were higher in PS/Fe$^{3+}$ process than that in PS/Fe$^{2+}$ process. The reason for this may be that larger amounts of HO• and SO$_4^-$• formed in PS/Fe$^{2+}$ process and caused more side radical reactions (Equation (8)), resulting in the formation of more other radicals.

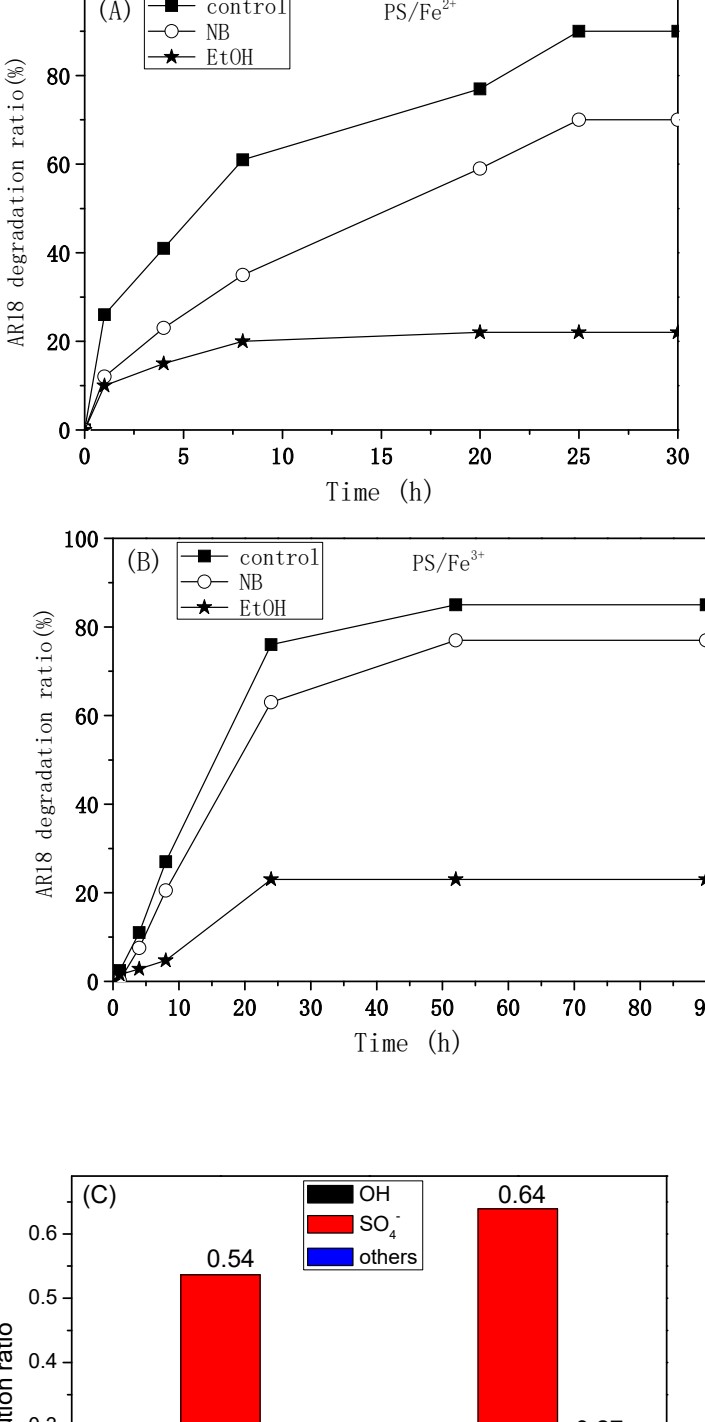

**Figure 8.** The AR18 degradation ratio with and without radical scavengers in PS/Fe$^{2+}$ (**A**) and PS/Fe$^{3+}$ processes (**B**), and the contribution of different oxidants for AR18 degradation in PS/Fe$^{2+}$ and PS/Fe$^{3+}$ processes (**C**) (initial pH = 3, $C_{Fe2+}$ = $C_{Fe3+}$ = 0.1 mM, $C_{PS}$ = 0.5 mM, $C_{AR0}$ = 300 mg/L, $C_{EtOH}$ = 500 mM, $C_{NB}$ = 500 mM).

## 3. Materials and Methods

### 3.1. Materials

AR18 was purchased from Jiaying Chemical Factory (Shanghai, China). The molecular formula of AR18 is $C_{20}H_{11}N_2Na_3O_{10}S_3$ and the maximum absorbance wavelength ($\lambda_{max}$) is 507 nm (Figure 9). Ferrous sulfate (FeSO$_4\bullet$7H$_2$O), ferric nitrate (Fe(NO$_3$)$_3\bullet$9H$_2$O), nitrobenzene (NB), and ethanol (EtOH) were obtained from Beijing Chemical Works, China. Dimethyl-1-pyrroline N-oxide (DOMP) was purchased from TCI (Shanghai, China) Development Co., Ltd.

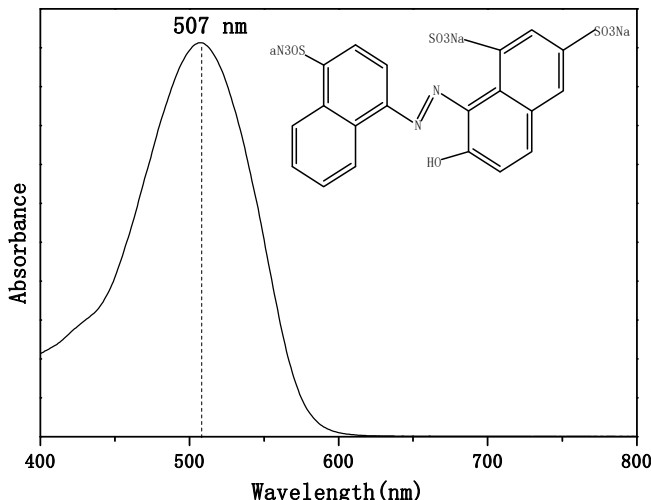

**Figure 9.** The molecular structure and absorbance spectrum of AR18.

### 3.2. Experimental Procedures

All experiments were carried out in a 250 mL flask at a temperature of 20 °C. The AR18 was dissolved in deionized water to prepare the simulated wastewater. Ferrous sulfate or ferric nitrate was added into the AR18 solution to attain a certain concentration of Fe$^{2+}$ (C$_{Fe2+}$) or Fe$^{3+}$ (C$_{Fe3+}$). The initial pH was adjusted by using sulfuric acid or sodium hydroxide with a pH meter (model PHS-25, Shanghai Precision & Scientific Instrument Co. LTD, Shanghai, China). The reaction was initialized by the addition of PS solution. Samples were taken from the flask by using a pipette at given intervals.

In the radical analysis experiments, the persulfate solution (5 mL, 6 mM) with spin trap agent (DMPO, 0.1 M) were activated by Fe$^{2+}$ or Fe$^{3+}$ (0.1 mM) for 7 min, and then the solution was sampled with a quartz capillary and analyzed immediately by an electron paramagnetic resonance spectrometer (EPR, Bruker EMX 10/12, Karlsruhe, Germany).

### 3.3. Analyses

The UV-VIS spectrum of AR18 was analyzed by using a UV-VIS spectrophotometer (UV-5200 PC, Shanghai Metash Instruments CO., LTD. Shanghai, China. The concentration of the AR18 was obtained by monitoring the absorbance at $\lambda_{max}$ and computing the concentration with a calibration curve. Total organic carbon (TOC) was determined by a TOC analyzer (multi N/C 2100, Analytik Jena, Jena, Germany). The electron paramagnetic resonance (EPR) spectra were obtained at room temperature with a microwave frequency of 9.856 GHz, microwave power of 2 mW, receiver gain of $6.32 \times 10^2$, modulation frequency of 100 kHz, modulation amplitude of 1.0 G, time constant of 0.01 ms, sweep time of 10.24 s, and sweep width of 50 G.

## 4. Conclusions

In this work, AR18 was successfully degraded by PS/Fe$^{2+}$ and PS/Fe$^{3+}$ processes, and the degradation ratio increased with the increasing concentrations of PS, Fe$^{2+}$, and Fe$^{3+}$ with an optimized pH of 3 and 3.3 for PS/Fe$^{2+}$ and PS/Fe$^{3+}$, respectively. According to the EPR study for radical formation, HO$\bullet$ and SO$_4^-\bullet$ formed in both PS/Fe$^{2+}$ and PS/Fe$^{3+}$ processes and played a dominant role in the degradation of AR18. The results of quenching experiments revealed that the contribution of HO$\bullet$ and SO$_4^-\bullet$ on AR18 degradation was higher in PS/Fe$^{3+}$ process than that in PS/Fe$^{2+}$ process, which can be attributed to the fewer side radical reactions in the process of PS/Fe$^{3+}$. As ferric salt is cheap and easy for storage, PS/Fe$^{3+}$ process would be a feasible method for azo dye wastewater treatment.

**Author Contributions:** Funding acquisition, L.Z.; investigation, X.L.; methodology, L.Y.; writing—original draft, X.L.; writing—review and editing, L.Z. All authors have read and agreed to the published version of the manuscript.

**Funding:** This research was funded by the Science and Technology Major Project of Shanxi Province (No. 20181101018), Bidding Project of Shanxi Province (20191101001), Youth Scientific and Technological Foundation of Shanxi Province (201901D211584), and the Key R&D Project of Shanxi Province (201703D111008).

**Conflicts of Interest:** The authors declare no conflicts of interest.

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
