# Peer review of "A Comparative Study on Oxidation of Acidic Red 18 by Persulfate with Ferrous and Ferric Ions"

_catalysts, doi:10.3390/catal10060698_

Round 1
Reviewer 1 Report
The topic of the manuscript is certainly of interest for Catalysts’readership. I recommend its publication after major revisions
- Some grammar and format mistakes occurred in the manuscript. E.g.
Line 14. “Sulfate radial” should be “Sulfate radical”
Line 25 “The present of azo dye” should be “The presence of azo dye”
Line 28 Advanced oxidation processes (AOPs) charactered by should be “ Advanced oxidation processes (AOPs) characterized by the”
Line 43-51 check all the apexes in the eq. 1-9
Line 94-96 : the sentence is not clear
- The authors investigated first the effect of initial pH on the efficiency of the oxidative process and they found the optimal pH values of 3.3 and 3 for PS/Fe2+ and PS/Fe3+ processes , respectively. Why did they carry out the successively experiments (Effect of PS concentration , Effect of Fe2+ or Fe3+ concentration) by using an initial pH value of 4.5?
- Line 129: the authors affirm that the low increment in AR18 degradation rate observed for the PS/Fe3+ system upon increasing the concentration of Fe3+ is due to the low concentration of reducing organic radicals . From eq. 4, however , the concentration of R depends on the concentration of PS , so I am wondering why in the PS/Fe3+system is such concentration lower than that of PS/Fe2+?
- Line 160:The low recycle rate of Fe would be a probable reason of the lower AR18 and TOC degradation rates in PS/Fe3+ process. The authors should explain what they intend for low recycle rate of Fe
- In fig 8 identify the labels B, C, and D
Author Response
Reviewer #1: The topic of the manuscript is certainly of interest for Catalysts’ readership. I recommend its publication after major revisions
- Some grammar and format mistakes occurred in the manuscript. E.g.
Line 14. “Sulfate radial” should be “Sulfate radical”
Line 25 “The present of azo dye” should be “The presence of azo dye”
Line 28 Advanced oxidation processes (AOPs) charactered by should be “ Advanced oxidation processes (AOPs) characterized by the”
Line 43-51: check all the apexes in the eq. 1-9
Line 94-96 : the sentence is not clear
Reply 1: We are sorry for the mistakes. We have checked and made corrections in the revision.
- The authors investigated first the effect of initial pH on the efficiency of the oxidative process and they found the optimal pH values of 3.3 and 3 for PS/Fe2+ and PS/Fe3+ processes, respectively. Why did they carry out the successively experiments (Effect of PS concentration, Effect of Fe2+ or Fe3+ concentration) by using an initial pH value of 4.5?
Reply 2: According to the investigation of the effect of initial pH (section 2.2), the AR18 degradation rate did not increase significantly with the decrease of initial pH in PS/Fe2+ and PS/Fe3+ processes when the initial pH was less than 3. The initial pH of AR18 solution (60 mg/L) was 4.5. Therefore, a suitable value in this study was in the range of 3-4.5. In order to minimize the operation cost, the initial pH was set at 4.5 in the other single-factor experiments.
- Line 129: the authors affirm that the low increment in AR18 degradation rate observed for the PS/Fe3+ system upon increasing the concentration of Fe3+ is due to the low concentration of reducing organic radicals. From eq. 4, however, the concentration of R depends on the concentration of PS , so I am wondering why in the PS/Fe3+system is such concentration lower than that of PS/Fe2+?
Reply 3: As we can know from Eq.4, the formation rate of R• depends on the concentration of radicals (HO• or SO4−•), which correlates to the concentration of both PS and Fe2+. In the early stage of PS/Fe2+ process, the higher concentration of Fe2+ leading to a higher formation rate of radicals and R•. Therefore, a higher concentration of reducing organic radicals were attained in PS/Fe2+ process with a same PS concentration as PS/Fe3+ process.
- Line 160: The low recycle rate of Fe would be a probable reason of the lower AR18 and TOC degradation rates in PS/Fe3+ process. The authors should explain what they intend for low recycle rate of Fe
Reply 4: Since iron catalytic process is easy to operate and has a high oxidation efficiency, Fe2+ is the most widely used transition metal ion for initializing radical chain reaction in PS activation process. The high reaction rates of Fe2+ and oxidants make PS/Fe2+ process transfer to PS/Fe3+ process rapidly. However, the catalytic oxidation process does not stop when the Fe2+ were completely consumed. Fe3+, the oxidative product of Fe2+, is also an effective catalysis for PS oxidation process as a result of the Fe recycle. As the prices of ferric salts are lower than that of ferrous salts, the reagent cost could be reduced by replacing ferrous salts with ferric salts in PS oxidation process. However, little study has been reported about the mechanism and the kinetic of contaminant oxidation by PS/Fe3+ process under different operation conditions. The role of Fe3+ on the PS activation and pollutions degradation is still remains enigmatic. Therefore, the PS activated by Fe2+ or Fe3+ were investigated in this study.
- In fig 8 identify the labels B, C, and D
Reply 5: We are sorry for the mistakes. We have checked and made corrections in the revision.
Reviewer 2 Report
The paper can be published after the following revisions:
1) it is not surprising that the initial pH of 3.3 and 3 was the optimal pH (please give more explanation about the influence of pH).
2) The concentration of the AR18 was obtained by monitoring the absorbance at λmax and computing the concentration with a calibration curve: please specify the value of λmax and show a typical absorbance spectrum of the selected dye.
Author Response
Reviewer #2: The paper can be published after the following revisions:
- It is not surprising that the initial pH of 3.3 and 3 was the optimal pH (please give more explanation about the influence of pH).
Reply 1: We thanks the reviewer for the suggestions and have elaborated the effect mechanism of the initial pH on PS activations in the revision (part 2.2) as following.
The residual AR18 concentration reached a valley value and the reaction kinetic constant reached the peak value with the initial pH of 3 in PS/Fe3+ process and pH of 3.3 in PS/Fe2+ process. A similar conclusion could be found in previous studies on H2O2/ Fe2+ systems [22]. Though Ferrous salts can be dissolved under a pH of 2-9, the ferric oxyhydroxides may be generate under a higher pH (>3.0) [32]. A higher pH would lead to a loss of soluble iron ions through the precipitation process, and result in a decrease of the AR18 degradation rate. It is necessary to balance the ferric oxyhydroxides formation and acid catalyzed PS breakdown.
- The concentration of the AR18 was obtained by monitoring the absorbance at λmax and computing the concentration with a calibration curve: please specify the value of λmax and show a typical absorbance spectrum of the selected dye.
Reply 2: The molecular structure and absorbance spectrum of AR18 have been added in the revision as Figure 9.
Reviewer 3 Report
This work reported an effective oxidation of acidic red with persulfate with ferrous and ferric ions. The results are attractive but both introduction and the experiment parts can be improved.
- For the introduction, the authors should carefully compare their methods to photocatalytic oxidation since the latter can be conducted in a near neutral environment and with a solar energy (e. g., J. Photochem. Photobio A 256 (2013) 7; 332 (2017) 457). For the environmental friendly and energy efficient viewpoint it is critical to identify the advantage and shortcoming of the current method.
- For the experiment, it is also important to compare the current data to photocatalytic oxide and elucidate the merits.
Author Response
Reviewer #3: This work reported an effective oxidation of acidic red with persulfate with ferrous and ferric ions. The results are attractive but both introduction and the experiment parts can be improved.
For the introduction, the authors should carefully compare their methods to photocatalytic oxidation since the latter can be conducted in a near neutral environment and with a solar energy (e. g., J. Photochem. Photobio A 256 (2013) 7; 332 (2017) 457). For the environmental friendly and energy efficient viewpoint it is critical to identify the advantage and shortcoming of the current method. For the experiment, it is also important to compare the current data to photocatalytic oxide and elucidate the merits.
Reply: We thanks the reviewer for the suggestions and have added some discussions about photocatalytic oxidation and photo-fenton in the revision. Photocatalytic oxidation with UV or solar light were effective method for wastewater treatment. Especially the solar photocatalytic oxidation is an energy-saving method. The relatively low energy utilization would be the main defect of photocatalytic oxidation process. Since iron catalytic process is easy to operate and cost-effective, iron catalyst is the most widely used catalyst for initializing radical chain reaction in H2O2 or PS activation process. Therefore, the PS activated by Fe2+ or Fe3+ were investigated in this study.
Round 2
Reviewer 1 Report
ok for publication